



# Realistic ice-shelf/ocean state estimates (RISE) of Antarctic basal melting and drivers

Benjamin K. Galton-Fenzi[1,2,3], Richard Porter-Smith[1,2], Sue Cook[2], Eva Cougnon[4], David E. Gwyther[5], Wilma G. C. Huneke[6], Madelaine G. Rosevear[7,3], Xylar Asay-Davis[8], Fabio Boeira Dias[9,3], Michael S. Dinniman[10], David Holland[11], Kazuya Kusahara[12], Kaitlin A. Naughten[13], Keith W. Nicholls[13], Charles Pelletier[14], Ole Richter[15,16], Hélène Seroussi[17], and Ralph Timmermann[16]

[1]Australian Antarctic Division, Kingston, Tasmania, Australia
[2]Australian Antarctic Program Partnership, Institute for Marine and Antarctic Studies, University of Tasmania, Australia
[3]Australian Centre for Excellence in Antarctic Science, University of Tasmania, Australia
[4]Integrated Marine Observing System, Hobart, Tasmania
[5]University of Queensland, Australia
[6]Australian Centre of Excellence for Climate Extremes, Australian National University, Canberra, Australia
[7]Melbourne University, Australia
[8]Los Alamos National Laboratory, USA
[9]University of New South Wales, Australia
[10]Old Dominion University, Virginia, USA
[11]New York University, USA
[12]Japan Agency for Marine-Earth Science and Technology, Japan
[13]British Antarctic Survey, Cambridge, UK
[14]Earth and Life Institute (ELI), UCLouvain, Louvain-la-Neuve, Belgium
[15]University of Rostock, Germany
[16]Alfred Wagner Institute, Germany
[17]Dartmouth College, New Hampshire, USA

**Correspondence:** Benjamin K. Galton-Fenzi (ben.galton-fenzi@aad.gov.au)

**Abstract.**

Societal adaptation to rising sea levels requires robust projections of the Antarctic Ice Sheet's retreat, particularly due to ocean-driven basal melting of its fringing ice shelves. Recent advances in ocean models that simulate ice-shelf melting offer an opportunity to reduce uncertainties in ice–ocean interactions. Here, we compare several community-contributed, circum-

Antarctic ocean simulations to highlight inter-model differences, evaluate agreement with satellite-derived melt rates, and examine underlying physical processes. All but one simulation use a melting formulation depending on both thermal driving ($T^\star$) and friction velocity ($u^\star$), which together represent the thermal and ocean current forcings at the ice–ocean interface. Simulated melt rates range from 650 to 1277 Gt year$^{-1}$ ($m = 0.45 - 0.91$ m year$^{-1}$), driven by variations in model resolution, parameterisations, and sub-ice shelf circulation. Freeze-to-melt ratios span 0.30 to 30.12 %, indicating large differences in

how refreezing is represented. The multi-model mean (MMM) produces an averaged melt rate of 0.60 m year$^{-1}$ from a net mass loss of 842.99 Gt year$^{-1}$ (876.03 Gt year$^{-1}$ melting and 33.05 Gt year$^{-1}$ refreezing), yielding a freeze-to-melt ratio of 3.92 %. We define a thermo-kinematic melt sensitivity, $\zeta = m/(T^\star u^\star) = 4.82 \times 10^{-5}$ °C$^{-1}$ for the MMM, with individual models spanning $2.85 \times 10^{-5}$ to $19.4 \times 10^{-5}$ °C$^{-1}$. Higher melt rates typically occur near grounding zones where both $T^\star$ and





$u^\star$ exert roughly equal influence. Because friction velocity is critical for turbulent heat exchange, ice-shelf melting must be
characterised by both ocean energetics and thermal forcing. Further work to standardise model setups and evaluation of results
against in situ observations and satellite data will be essential for increasing model accuracy, reducing uncertainties, to improve
our understanding of ice-shelf–ocean interactions and refine sea-level rise predictions.

## 1   Introduction

Societal adaptation to rising sea levels needs to be informed by how retreat of the Antarctic Ice Sheet will occur under a future
warming climate and contribute to global sea level changes. One of the largest uncertainties in projections of Antarctic ice
sheet evolution is how much ocean-driven melting of the ice shelves fringing the Antarctic Ice Sheet is presently occurring
and how much is expected for the future (Seroussi et al., 2023). Melting of the Antarctic floating ice shelves by the ocean and
iceberg calving are the two main processes driving the mass loss of the Antarctic Ice Sheet at about the same rate (e.g. Greene
et al., 2022; Liu et al., 2015; Depoorter et al., 2013; Rignot et al., 2013, 2019).

Ocean-driven mass loss from the West Antarctic Ice Sheet is accelerating (Shepherd et al., 2018; Rignot et al., 2019; Schröder
et al., 2019; Sasgen et al., 2019) and has become a region of intense scientific scrutiny. Meanwhile, East Antarctica also
experiences ocean-driven mass loss, albeit to a lesser degree but with significant spatial variability, underscoring the need to
study all areas to fully understand the dynamics of the Antarctic Ice Sheet. It remains unclear how much ocean-driven melting
contributes to overall ice-sheet ablation and whether basal melt is more significant than previously assumed or has increased
in recent decades (e.g., Paolo et al., 2015). Moreover, determining the spatial distribution of basal melt rates is complicated
by heterogeneous ocean circulation processes that differ across Antarctica (Smith et al., 2020; Adusumilli et al., 2020). These
complexities must be addressed to accurately project future ice-sheet stability and its implications for global sea-level rise.

Increased basal melting can lead to thinning of the ice shelves, reducing buttressing and increased flow of ice from the
continent into the oceans (Pritchard et al., 2012). Otherwise confined ice shelves have their discharge speed reduced from the
mechanical sidewall friction (Thomas et al., 1979). The reduction in buttressing affects the grounded ice shelves and causes
the accelerated flow of tributary glaciers (Schoof, 2007). Therefore, basal ice melt not only directly causes mass loss and ice
thickness changes but also contributes to ice stream dynamics (Gagliardini et al., 2010). Thereby, understanding the holistic
magnitude and spatial distribution of basal ice melt is crucial not only to estimate ocean-induced melt itself but also to better
assess interconnected processes in relation to calving and surface melting, and in understanding and assessing current (Gwyther
et al., 2020b) and future mass loss from Antarctica. It is therefore a critical metric for predicting future ice sheet vulnerability.

In addition to contributing to sea level rise, basal meltwater from the Antarctic Ice Sheet plays an important role in several
key climate processes. The influx of fresh meltwater influences the formation of Antarctic Bottom Water (AABW), a critical
component of the global thermohaline circulation that drives deep ocean overturning (Chen et al., 2023). Changes in AABW



formation can alter the global heat distribution and affect climate patterns worldwide (Bennetts et al., 2024). Basal meltwater
also impacts the dynamics of coastal sea ice dynamics by modifying the salinity and temperature of nearshore waters, which
can lead to changes in sea ice extent and thickness (Bintanja et al., 2013). These impacts on sea ice have further implications
for marine ecosystems, modifying light penetration, phytoplankton growth, and altering nutrient distributions (Constable et al.,
2014). Understanding how basal meltwater is created and mixed into the oceans is important for projecting the future behavior
of the Antarctic environment and connection to the global climate system.

Although recent progress in cryosphere and ocean research has improved our knowledge of ocean-driven ice-shelf melting,
effectively addressing its scale and complexity calls for comprehensive, internationally coordinated efforts spanning diverse
methods and research programs (e.g., Gwyther, 2018; Cook et al., 2022). Investigations targeting Antarctica's most vulnerable
regions, as well as the underlying processes and feedbacks that drive melting and link to the global climate system, remain a key
focus (Gwyther et al., 2018). A central priority across climate and cryosphere communities (e.g., IUGG, WCRP, SCAR) is to
reduce uncertainties in projections of future Antarctic Ice Sheet evolution. Internationally coordinated mass-budget estimates,
through both observation-based (Shepherd et al., 2018; Otosaka et al., 2023) and modeling-focused (Nowicki et al., 2020;
Jourdain et al., 2020; Seroussi et al., 2020) efforts, must incorporate accurate representations of basal melting and its likely
changes. Multiple initiatives now tackle ice–sheet/ocean interactions using state-of-the-art methodologies that differ in both
approach and outcomes. For example, MISOMIP (Asay-Davis et al., 2016) and MISOMIP2 (De Rydt et al., 2024) center on
idealized and more regional ice–ocean modeling, respectively, underscoring the diversity and complexity of ongoing research
in this field.

Despite notable progress in satellite-derived assessments of Antarctic ice-sheet mass loss (e.g., Shepherd et al., 2018; Rignot
et al., 2019; Otosaka et al., 2023), several limitations remain. Data coverage can be restricted by orbital geometry and sensor
resolution (McMillan et al., 2014; Paolo et al., 2015), while corrections for grounding-zone flexure (Brunt et al., 2010) and
firn-layer thickness (Kuipers Munneke et al., 2013; Ligtenberg et al., 2014) may introduce substantial uncertainties. Detecting
subtle changes in basal melting beneath thick ice shelves also proves difficult (Khazendar et al., 2016; Adusumilli et al., 2020),
and reconciling satellite measurements with in situ observations is vital for refining existing estimates (Cook et al., 2022).
Although on-ice measurements and remote sensing products capture large-scale signals of ice-mass change, these methods
often cannot reveal the causal oceanic processes in data-sparse environments. In such regions, high-resolution ocean models
offer a crucial perspective on the mechanisms driving basal melt (Favier et al., 2019; Jourdain et al., 2020), thus improving
our understanding of how observed mass loss arises. Combining advanced modeling approaches with enhanced observational
networks will be key to capturing the complexities of Antarctic ice-shelf melting and producing more reliable predictions of
sea level rise (Seroussi et al., 2020; Galton-Fenzi et al., in-press).

However there has been a significant lack of comparative studies of why ice-shelf/ocean models often produce large differ-
ences in simulated rates of basal ice melt, as compared with satellite estimates (see Figure 6 in  Richter et al., 2022). From
idealised experiments, the divergence is in part due to unique parameterisations, model-specific numerics and discretisation,
and different choices of boundary forcing (Holland et al., 2003; Hunter, 2006; Asay-Davis et al., 2016; Gwyther et al., 2020a).
Basal melt rates simulated for all parts of the Antarctic coastline differ between different models. Comparing these results al-





lows us to constrain both present and future ocean-driven impacts on Antarctica, and provide much needed evaluation with both satellite-derived and in situ estimates of ice shelf basal melt rates. This comparison work and evaluation will also be valuable to observations based programs by informing them on potential areas of specific interests for future field programs. Therefore, the project contributes to the understanding of ocean-driven melting of Antarctic ice shelves and thus to the understanding of sea level rise contributed by the Antarctic Ice Sheet in a changing climate.

This paper is an output from the Realistic Ice Shelf-Ocean Estimates (RISE) Project. Unlike traditional Model Intercomparison Projects, RISE does not prescribe an experimental design; instead, it aims to compare existing Antarctic ocean and ice-shelf model outputs with basal melt rate estimates across all Antarctic ice shelves. The objective is rather to compare existing Antarctic Ocean/ice-shelf models outputs with estimates of basal melt rates for all Antarctic ice shelves. RISE was designed to leverage the rapid advancement in modelling and coordinate the outputs with observational programs, such as

remotely-sensed observational program (e.g., ICESat; Paolo et al., 2015, 2018) and ground-based observation programs (e.g., the coordinated use of ApRES as part of NECKLACE a SOOS endorsed activity - https://necklaceproject.com/). This study is intended as an initial overview paper, providing results that are crucial for the community, to support climate research and modeling efforts.

    Given the high cost of developing and producing high fidelity simulations of ice-ocean melting, we used all available circum-

Antarctic ocean/ice-shelf simulations. Numerical models serve as indispensable tools for simulating the dynamic processes governing Antarctic ice melt. Yet, individual models often exhibit biases and uncertainties, stemming from simplifications of complex physical phenomena including poor information and fiedility of the forcing conditions. The utilisation of a multi-model mean has emerged as a promising strategy to mitigate these limitations and enhance predictive accuracy (e.g., Tebaldi and Knutti, 2007; Gleckler et al., 2008; Smith et al., 2009; Knutti and Sedláček, 2013). We have included an analysis of melt

rate data from nine simulations to produce a multi-model mean in this study, and compare them to satellite-derived measurements of ice shelf basal melt. The approach includes comparing each model's outputs to the -derived product; describing the variance between the models to examine where they agree or disagree most strongly; and to identify any systematic differences (i.e., whether the ice-ocean models systematically over- or under-predict melt at the grounding line or fail to reproduce refreezing). As well as providing a useful comparison between different models, this study will also guide the direction of future

observations on and beneath ice shelves, thereby integrating ice sheet/ice shelf-ocean observations and modelling.

## 2    Methods

We use output from nine circum-Antarctic ice-shelf/ocean simulations estimating ice-shelf basal melt rate for all Antarctic ice shelves. The approach used here allowed any available model to contribute to the comparison study without the need for extensive standardization. This approach facilitates the use of available models produced by various international groups, but

meant the contributing models used a range of both forcing and parameters, and numerical discretisation methods for their simulations and as such we refer the reader to the appropriate reference for each model (Table 1). The only requirement was that models needed to span Antarctica. We note the models were each run for their own time period.





**Table 1.** Summary of models used in this study, including evaluation data from satellite-based measurements. Details for each model can be found in the corresponding primary reference for each model. $u^\star$ shown here is that used within each model to determine the melt rate. The grid resolutions for E302, FESH and FESL are average values only as these three models use finite element horizontal discritisation and therefore have varied resolution. Further details for each model can be found in the supplementary material.

| Identifier | Model Name | Reference | Averaging period | Resolution | $u^{\star 2}$ (m s$^{-1}$) |
|---|---|---|---|---|---|
| MMM | Multi Model Mean | This study | see text | 2 km | - |
| COCO | COCO | Kusahara (2021) | 1979-2018 | $1/5°\times1/5°\cos(\phi)$ | [‡]$7.225 \times 10^{-3}$ |
| DINN | ROMS3.6 | Dinniman et al. (2020) | 2010 | 5 km | $6.0 \times 10^{-3}\overline{uv}_m^2$ |
| E302 | E3SM/MPAS-O | Comeau et al. (2022)[1] | 150 years | ~10-30 km | [†]$6 \times 10^{-3}(\overline{uv}_m^2 + u_{\text{tides}}^2)$ |
| FESH | FESOM | Naughten et al. (2018) | 1992-2016 | ~3-10 km | $2.0 \times 10^{-3}\overline{uv}_m^2$ |
| FESL | FESOM | Naughten et al. (2018) | 1992-2016 | ~10-30 km | $2.0 \times 10^{-3}\overline{uv}_m^2$ |
| METR | MetROMS | Naughten et al. (2018) | 1992-2016 | $1/4°$ | $3.0 \times 10^{-3}\overline{uv}_m^2$ |
| NE01 | NEMO_bmbath | Pelletier et al. (2022)[2] | 1979-2018 | $1/4°\times1/4°\cos(\phi)$ | [§]$1.0 \times 10^{-3}(\overline{uv}_m^2 + min(TKE))$ |
| NE02 | NEMO_fETv171 | Pelletier et al. (2022)[2] | 1979-2018 | $1/4°\times1/4°\cos(\phi)$ | [§]$1.0 \times 10^{-3}(\overline{uv}_m^2 + min(TKE))$ |
| RICH[3] | WAOM-ROMS3.5 | Richter et al. (2022) | 2007 | 2 km | $5.0 \times 10^{-3}\overline{uv}_m^2$ |
| SATT | Satellite | Adusumilli et al. (2020) | 1994-2018 | 10 km | - |

[‡] COCO was the only model to use fixed exchange coefficients that determined the melt rate as a function of the thermal driving only. The $u^\star$ provided here is that which yields the fixed exchange coefficients used.

[†] E3O2 includes a fixed velocity in the melt rate parameterisation to include the influence of tides, $u_{\text{tides}} = 5.0 \times 10^{-2}$ m s$^{-1}$.

[S] NEMO includes in the calculation of $u^\star$ a minimum Turbulent Kinetic Energy for the surface and bottom boundaries of the ocean model, $min(TKE) = 1.0 \times 10^{-4}$ m$^2$ s$^{-2}$, equivalent to a background current with a velocity of $1.0 \times 10^{-2}$ m s$^{-1}$.

[1] Simulations used a configuration that is unpublished.

[2] The stand alone ice-shelf/ocean model component was used from this study under two configurations that are unpublished.

[3] RICH is the only model to explicitly include tides, as the 10 major tidal constituents.

[4] Sea ice fluxes are prescribed, combining reanalysis products (see appendix) with satellite estimates of sea ice production (Tamura et al., 2011).

The nine simulations contributed here were produced from five different models: COCO (1), NEMO (2), FESOM (2), ROMS (3), E3SM/MPAS-O (1) - see Table 1 for the main characteristics of the models. NEMO was run with two different estimates of the bathymetry, FESOM was run at two different resolutions, and three ROMS applications were run at different resolutions, with and without tides and a choice of either a sea ice model or prescribed surface fluxes. We compare results against the most recent satellite-derived estimates of basal melt rates from satellite altimetry acquired during 2003–2008 (Adusumilli et al., 2020). The outputs from each simulation were averaged over time (see Table 1 descriptions) to produce a generic file of available variables and standardised units. The generic files were subsequently used within a GIS platform, R, and Matlab for further analysis and plotting. All models provided the fundamental variables needed for comparison including the basal melt rate, ocean temperature and salinity. However, not all models provided the parameters used in the three-equation model ($T^\star$ and $u^\star$), which subsequently needed to be estimated for those that did not (see discussion below).



Each of the contributing simulations was done over different time periods. The resulting contribution from each model to RISE is as a time-mean, representing the mean ocean state over the averaging period of each simulation period. These models

were then used to produce an ensemble multi-model mean (MMM) that we use for futher analysis. In the calculation of the MMM, we did not attempt to bias between different averaging periods. However, most of the models include the period from the early 1990s to the mid-2010s (Table 1), and share a common time period that is centered around the early 2000s, with 2004 the most common year, approximately similar to the average period of the satellite observations of 2006. The MMM best represents the mean ocean (and hence basal melting) state of the median year of 2004, but it is still representative of the average

of simulations from 1990 to the late 2010s, given most of the models cover this period.

Parameterisations are necessary to predict melting and freezing in ice-ocean models at the scale used here, since the relevant scales of motion are not resolved. The three-equation melt parameterization (Holland and Jenkins, 1999a) uses ocean conditions in the mixed layer below the ice for the temperature, $T_m$, salinity, $S_m$ and currents $u_m$, to predict interface temperature, salinity and melt rate $(T_b, S_b, m)$ using equations that encapsulate turbulent transfer across the ice-ocean boundary layer by relating

$u_m$ to the friction velocity, $u^\star$. Estimating the mixed-layer properties is handled by each model, which evolves the three-dimensional circulation within the ice shelf cavity. In practise, $T, S$, and $u$ are usually taken at the grid cell closest to the ice-ocean interface, or averaged over some thickness near the upper layer (e.g. Gwyther et al., 2020a). The COCO model is the only model contributed that uses a parametrization that depend on $T_m$ and $S_m$ but not depend on current speed $u_m$ and instead use constant exchange velocities.

## 140  2.1  Estimation of the thermal driving, $T^\star$, and friction velocity, $u^\star$

The quantities known as the thermal driving, $T^\star$, and the friction velocity, $u^\star$, are the dominant drivers of basal ice melt and freeze within the framework of the three-equation parameterisation. The thermal driving $T^\star = T_m - T_f$ is the elevation of the local temperature in the mixed layer $T_m$ above the *in situ* freezing temperature $T_f$ at the local salinity and pressure, and indicates how much heat is available to melt the ice.

Since not all of the RISE models supplied $u^\star$ and $T^\star$, it was sometimes necessary to estimate these quantities from the provided model output. All quantities were converted to Absolute Salinity to facilitate a comparison. Calculating $T^\star$ typically involves either sampling temperature and salinity in the top model cell, or averaging across multiple cells within 2 to 40 m of the ice (Gwyther et al., 2020b). However, for this experiment, the choice was made to sample and calculate it from the temperature and salinity of the top model layer, directly under the ice-draft, converted to Absolute Salinity. Thermal

Driving, $T^\star = T_m - T_f(S_A, p)$, where $S_A$ is the absolute salinity (g kg$^{-1}$), $p$ is the pressure (dBar). To calculate $T^*$, model output potential temperature ($\theta$) was first converted to *in situ* ambient temperature (ITS-90) to obtain $T_m$. The *in situ* freezing temperature ($T_f$) was then computed by a modified Newton-Raphson iteration (McDougall and Wotherspoon, 2014) using model estimates of salinity (g kg$^{-1}$), and pressure (dbar) in the top ocean layer (McDougall and Barker, 2011; McDougall et al., 2014).





The friction velocity is the turbulent velocity scale for the ice-ocean boundary layer, and depends on both the strength of the free-stream flow, and the roughness of the ice-ocean interface. For models that did not supply $u^\star$, it was calculated as:

$$u^{\star 2} = c_d \overline{uv}_m^2, \tag{1}$$

where $\overline{uv}_m = \sqrt{u_m^2 + v_m^2}$ (m s$^{-1}$) is the speed of the ocean in the upper ocean layer, $u$ is the zonal velocity (m s$^{-1}$), $v$ is the meridional velocity (m s$^{-1}$), and $c_d$ is the drag coefficient (see Table 1):

We compare the local melt rate in each model grid cell against the "thermo-kinematic forcing" ($T^\star u^\star$), to investigate the overall melt sensitivity to the local ocean conditions, and to facilitate evaluation across the contributing models and the MMM. The thermo-kinematic forcing metric is an approximation, since the dependence of melt on these parameters in the three-equation parameterizations used in the models is non-linear. Assuming $m \propto (T^\star u^\star)$ is similar to using a two-equation parameterization, where the influence of salinity is not explicitly included and the ratio of heat to salt transfer to the ice-ocean interface

is kept constant (e.g., see discussion in Jenkins et al., 2010). The thermo-kinematic forcing also does not include the effect of conductive heat flux into the ice shelf. This term is treated differently between models and is only expected to affect melt rates by $1 - 10\%$ (Gwyther et al., 2012; Holland and Jenkins, 1999b).

The thermo-kinematic forcing as we define it here is a local quantity, calculated at each grid cell. Therefore, it differs significantly from empirical relationships between melt and ocean temperature found in previous studies which instead considered

the relationship between cavity average melt rates and mixed layer or continental shelf temperatures, since this latter approach includes feedbacks between melt and buoyancy-driven overturning via the effect of circulation enhancing $u^\star$ (e.g. Holland et al., 2008b; Burgard et al., 2022).

## 2.2 Remapping approach to a common grid

One of the challenges of combining model outputs is to reconcile the various irregular mesh points from each of the models

to a common grid. This was overcome by using the concept of Voronoi tessellation (see supplementary material). A Voronoi tessellation is a partition of a plane into regions, where each region consists of all points closest to a specific location. Each region is closer to the region's point than to the point of any other region. Thereby, all areas of the plane are divided up into areas closest to each location. These are called Voronoi cells, also known as Thiessen polygons (Burrough et al., 2015; Longley et al., 2005; Sen, 2016). Once the correct proximal polygonal areas were established for each set of irregular points, the datasets

have been converted to regular raster grids for comparison and analysis allowing each model output to be directly compared to each other (see Supplementary material for details).

For computational efficiency, output was excluded from each of the simulations north of 50 degrees South and the longitude was adjusted from 0–360 to -180–180 degrees East where needed. These quantities were then projected from geographic coordinates (EPSG: 4326) into polar stereographic (EPSG: 3031) coordinates (Snyder, 1987; Snyder and Voxland, 1989) for

analysis. To overcome the differing scales and irregular sized meshes, points and associated data were extracted according to ice shelf boundaries defined by the Antarctic Boundaries for the International Polar Year 2007-2009 from Satellite Radar,





**Table 2.** Summary of key basal mass loss attributes for the multi-model mean (first line), all contributing simulations, and satellite-based estimates (last line). The last column shows the percentage ratio of freeze to melt. Values in parenthesis are the individual published model estimates. Values for the seven main ice shelves are presented in the supplementary information.

| Identifier | Melt rate (m year$^{-1}$) | Mass loss (Gt year$^{-1}$)$^\dagger$ | | | Freeze/Melt $\times 100$ (%) |
|---|---|---|---|---|---|
| | | Net | Melt | Freeze | |
| MMM | 0.60 | 843 | 876.03 | 33.05 | 3.92 |
| COCO | 0.76 | 1070 (1284) | 1107.64 | 37.71 | 3.52 |
| DINN | 0.44 | 611 | 700.00 (826) | 88.73 | 14.51 |
| E302 | 0.91 | 1274 | 1277.47 | 3.83 | 0.30 |
| FESH | 0.44 | 622 (739) | 809.32 | 187.32 | 30.12 |
| FESL | 0.37 | 523 (586) | 680.68 | 157.55 | 30.12 |
| METR | 0.45 | 636 (642) | 650.30 | 14.25 | 2.24 |
| NE01 | 0.83 | 1159 | 1198.71 | 39.64 | 3.42 |
| NE02 | 0.51 | 718 | 744.14 | 25.92 | 3.61 |
| RICH | 0.70 | 973 (1209) | 1025.38 | 51.99 | 5.34 |
| SATT | 0.88 | 1184 (1260) | 1407.70 | 223.28 | 18.85 |

$^\dagger$ Mean melt rates and mass loss rates are in units of freshwater mass per time assuming a freshwater density $\rho = 1000$ kg m$^{-3}$, where 1 Gt (Gigatonne) $= 1 \times 10^{12}$ kg

Version 2 (Mouginot, 2017) acquired from the the United States National Snow and Ice Data Center (NSIDC). The data used included coastline, islands, ice shelves (including naming convention) and associated grounding lines.

In this study, we analyze outputs from several ice-ocean numerical models estimating basal ice melt. However, each model often uses unique parameterisations and computational methodologies, resulting in sometimes diverse predictions of ice melt dynamics. We examine the distribution of model predictions and compare multi-model averaging and how this can mitigate individual biases and uncertainties. The analysis focuses on the multi-model mean for the whole of Antarctica and seven individual ice shelves - Amery, Fimbul, Larsen C, Ronne-Filchner, Ross, Thwaites and Totten (see Fig 1 for locations ) - and then explores the distributions of the melt rate and drivers across all models. These ice shelf cavities were chosen as being representative of different types and scales of ice shelves from around Antarctica and were included in all the simulations. Results for each of the seven ice shelves are shown in the supplementary material. The relationships between melting and the thermal driving is then discussed.

## 3 Results

An aspect of our methodology, with the remapping of melt only over regions defined by a common boundary, rather than the ice shelf areas on each model's native grid, yields lower total melt estimates than previous studies (Table 2). For example, values





published for the RICH simulation show about $1209 \, \text{Gt year}^{-1}$ of melt, whereas our approach produces only $973 \, \text{Gt year}^{-1}$, however we note Richter et al. (2022) obtains similar answer to us when averaged over the MEASURES areas. Similar discrepancies appear for other models, with ratios between published melt rates ranging from 1.01 to 1.2 times larger tan the results we present here. We also observe that the difference is more pronounced for high-resolution models, suggesting that signifi-

cant basal melting may occur near, or even slightly beyond, the NSIDC boundaries used here. In such areas, small changes in domain definitions can omit regions with relatively intense melt, contributing to lower total estimates.

The mismatch in spatial boundaries underscores why modeled melt totals may differ from satellite-derived products, such as those in Adusumilli et al. (2020). If the satellite data are also integrated over slightly different shelf extents or epochs, direct comparisons become problematic and may give the impression that model-based melting is systematically underestimated.

Moreover, high-resolution models may resolve steep gradients in melt rates near grounding lines or ice-shelf fronts that remain partially outside the NSIDC definitions. This pattern emphasizes how finer resolution can reveal intense local melting in narrow regions, thereby substantially affecting global estimates. We thus caution that using any fixed and common boundary for intercomparison could bias results downward for certain models, especially if their native grids extend beyond these predefined areas.

The ensemble Multi-Model Mean (MMM) melt rate for all ice shelves shows high melting in the Amundsen and Bellingshausen sectors (Figure 1a). East Antarctica has some regions of moderate melting, including Cook, Totten, the eastern Shackleton, Fimbulisen, Lazarev and Borchgrevink ice shelves. The Amery Ice Shelf also exhibits relatively high melting at its deepest portion. The Ronne-Filchner and Larsen C ice shelves have some elevated melting at their very deepest extents. Like the Ross, the Ronne Ice Shelf exhibits significantly large patches of refreezing. For more spatial detail and melt rates for

each contributing model, see supplementary material. The spatial distribution of melt rates is in qualitative agreement with the satellite-inferred mean melt rates (see supplementary material); the region with the largest difference is below the Amery Ice Shelf, where the model ensemble is not simulating enough refreezing, and shows a large disagreement between models, unlike the other two large embayed ice shelves - the Ronne-Filchner and Ross. This may be caused by the inability of most models to capture refreezing dominated by frazil accumulation – a process known to be important below the Amery ice shelf

(Galton-Fenzi et al., 2012).

The ensemble standard deviation in melt rate for all ice shelves (Figure 1b) shows the variability between ensemble member models, and hence represents the model spread. Interestingly, the Ronne and Ross ice shelves show relatively lower standard deviation, except at teh ice shelf fronts, perhaps due to most models producing low melting and refreezing that is known for these large ice shelves. In comparison the Amery Ice Shelf, which is also known to have high areas of refreezing, and

almost all other ice shelves have high levels of variance between model members, especially the relatively warm cavity ice shelves (Bellingshausen–Amundsen sectors, Totten and Moscow University ice shelves). Our analysis reveals that individual model predictions often exhibit distinctive distributions, reflecting inherent biases and uncertainties. However, when aggregated through multi-model averaging, these tend to converge towards being normally distributed. This convergence phenomenon demonstrates an advantage of using multi-model averaging in reconciling disparate model outputs and refining predictions of

Antarctic ice melt.



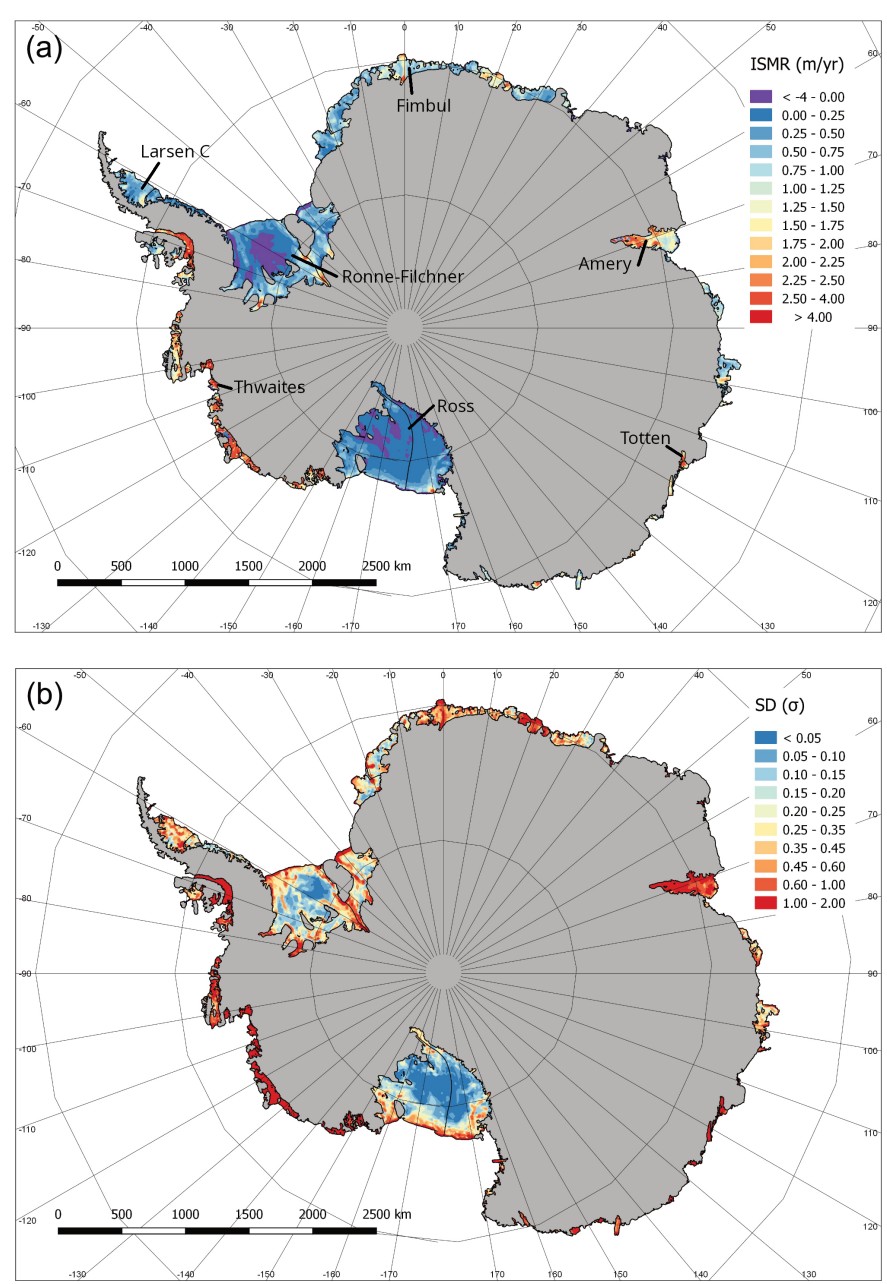

**Figure 1.** Multi-model mean of ice shelf (a) melt rate, and (b) standard deviation in melt rate.

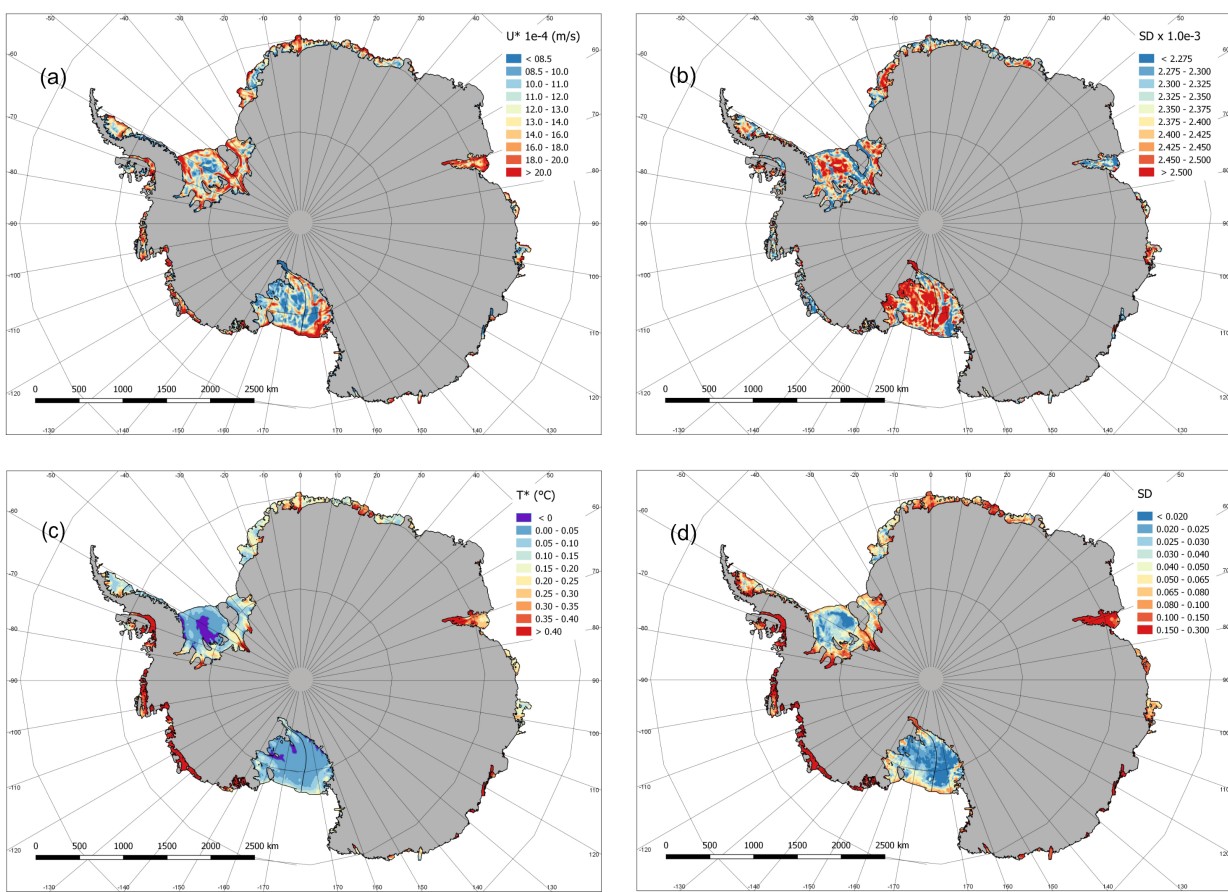

**Figure 2.** Multi-model mean of (a) friction velocity ($u^\star$), (b) standard deviation in $u^\star$, (c) thermal driving ($T^\star$), and (d) standard deviation in thermal driving.



As melting is driven by both the amount of heat and the degree to which this can be supplied to the ice-ocean interface, melting is a function of both $u^\star$ (Figure 2a) and $T^\star$ (Figure 2c). While the qualitative agreement between $T^\star$ and melt (Figure 1a) is evident, the contribution of $u^\star$ is equally important. Bands of high $u^\star$ typically illustrate strong flow regions, such as pathways of strong inflow into ice shelf cavities (e.g. adjacent to Ross Island under the Ross Ice Shelf), glacial meltwater adjacent to the topography or regions of deep ice, or ice shelf fronts.

Given the broad focus of this study, we concentrate further analysis on integral basal melt and driving parameters; these integrated quantities combine together all of the physical processes that must occur to produce melting. We do not compare with observationally derived estimates of water mass properties as these are incredibly sparse in space and time for the sub-ice-shelf cavity regions, or with other products on the continental shelf, given the widely varying epoch times used. Furthermore, most of the contributing models have already been individually evaluated elsewhere.

All of the individual models yield lower melt rates than the satellite-derived estimate, resulting in the MMM of about 0.6 m year$^{-1}$ compared with the satellite estimate of 0.88 m year$^{-1}$. Some models produce melt rates as high as 0.91 m year$^{-1}$, while others remain as low as 0.37 m year$^{-1}$, highlighting a substantial spread in simulated melting conditions. The MMM suggests a net ocean-induced mass loss from Antarctica of approximately 876 Gt year$^{-1}$ due to melting and a freeze-to-melt ratio of about 3.92%, whereas the satellite-based estimates indicate 1407 Gt year$^{-1}$ from melting and a significantly higher freeze-to-melt ratio of 18.85%. Additionally, the mean melt rates tend to be higher than the median for both the models and the satellite estimates, suggesting that a limited number of regions with intense melting skew the averages. Discrepancies likely arise from differences in the areas used to compute melt rates, the epoch of the datasets, and the inherent challenges of mapping and comparing model outputs against observations. Additionally, previous studies indicate that satellite-derived products may overestimate both melting and refreezing, reinforcing the importance of considering methodological uncertainties and temporal coverage when evaluating model performance.

We use violin plots to display both the central summary statistics and the full probability density of a dataset by combining elements of a box plot with a kernel density estimate. The statistical distribution of melt for each model (Figure 3a) shows relatively high agreement between each model mean (red dot) and median (blue dot) and the sum of the quantities (labeled MMM). The satellite estimate of the mean melt ($\sim$0.8 m year$^{-1}$) is higher than any individual model, although the satellite-inferred median melt is close to the MMM. This may highlight that the satellite product is suggests smaller regions of higher melting, which affect the mean but leave the median unchanged.

For ocean currents (Figure 3b) and ocean temperature (Figure 3c) at the ice base, there is broad agreement between most of the models, with some exceptions. For example, E302 tends to simulate lower currents but warmer mean conditions; FESOM (FESH and FESL) and COCO generally simulate faster currents. Many models also display a double peak in the temperature distribution, representing two significant constraints due to the freezing point temperature dependence at the surface of the ocean and at the mean ice draft depth. The greatest difference between models is reflected in the sub-ice salinity (Figure 3d and also see the maps in the supplemental information), for which we currently lack a satisfactory explanation. Possible contributing factors include differences in boundary conditions, parameterisations of sea-ice and freshwater fluxes. Additional observational data and dedicated sensitivity analyses are needed to pinpoint the underlying causes of this variability and to



**Figure 3.** Violin plots of Antarctica-wide (a) mean melt rate, (b) current speed, (c) temperature, and (d) salinity, used in the calculations of the melt rate. Satellite estimated melt rates (see Adusumilli et al., 2020), are included in (a). Mean (red) and median (blue) values are indicated by circles for each model or dataset.



improve the representation of salinity-driven processes. Spatial maps of temperature, salinity, and melt for each of the seven focus ice shelves are provided in the supplemental information.

The spatial distribution of temperature and salinity for the MMM around Antarctica over the continental shelf shows coherent regional patterns of distinct water masses (Figure 4). Off-shelf forcing of warm, salty Circumpolar Deep Water (CDW)—particularly evident in the Amundsen sector—interacts with the local production of Dense Shelf Water (DSW), which is colder and saltier due to sea ice formation and air-sea fluxes, and Ice Shelf Water (ISW). While numerous continental shelf regions are relatively fresh and cooler near the coast, pockets of saltier CDW intrusion occur, as observed near Thwaites Glacier in West Antarctica. This area experiences significant melting, which partially freshens and cools the water column, yet remains dominated by intrusions of warm, salty CDW. Consequently, glacial meltwater released into this environment gives rise to localized freshening and some cooling adjacent to the coast.

The large ice shelves of Ross and Ronne-Filchner are dominated by relatively cold and salty Dense Shelf Water (DSW), likely driven by high air-sea-ice production outside the ice shelf cavities, whereas waters beneath the Amery Ice Shelf are comparatively warmer and fresher. Along the East Antarctic continental shelf, from just east of the Ronne-Filchner Ice Shelf to the Amery Ice Shelf, conditions remain fresh and relatively cool; however, from the Amery Ice Shelf around to the Ross Ice Shelf, the region is influenced by warm, salty Circumpolar Deep Water (CDW), which appears even saltier than waters in the west—possibly due to enhanced sea ice production in local polynyas (Figure 4).

Melting can be approximated as a linear function of friction velocity ($u^\star$) and thermal driving ($T^\star$) (Figure 5), which we refer to as the 'thermo-kinematic melt sensitivity' $\zeta = m/(T^\star u^\star)$, where $\zeta = 4.82 \times 10^{-5}$ $^\circ$C$^{-1}$ for the MMM. As expected, using this approach produces a good fit across most models, with $r^2$ values ranging from 0.72 to 0.98 and an ensemble mean relationship of $r^2 = 0.69$ (Table 3). It is expected that the averaging used to produce the MMM would result in a lower $r^2$. Many model outputs cluster at $T^\star$ values less than 0.005 $^\circ$C m s$^{-1}$, and melt rates less than $2 \times 10^{-7}$ m s$^{-1}$ (approximately 6.3 m year$^{-1}$), while the highest melt rates can reach around $1.7 \times 10^{-6}$ m s$^{-1}$ (about 54 m year$^{-1}$).

NEMO produces the steepest relationship with NE01 and NE02 with the lowest correlation ($\zeta = 18.1 - 18.4 \times 10^{-5}$ $^\circ$C$^{-1}$, $r^2 = 0.68$), likely due to the included minimum turbulent kinetic energy value specified in the model (see Table 1) that would both increase the melting (as with E3O2 which includes a tidal velocity only in the calculation of $u^\star$) and the overall ocean energetics, which would further enhancing melting. Conversely, COCO, with fixed exchange coefficients in the melting parameterisation (therefore $u^\star$ is constant), produces the shallowest slope and the highest correlation ($\zeta = 2.85 \times 10^{-5}$ $^\circ$C$^{-1}$, $r^2 = 0.98$). METR also produces a high correlation ($r^2 = 0.98$) although with a steeper slope ($\zeta = 7.73 \times 10^{-5}$ $^\circ$C$^{-1}$). At high melt rates and values of $T^\star u^\star$ most of the models slopes are steeper than the MMM, highlighting the majority of the correlation is produced from lower values of $m$ and $T^\star u^\star$ (not shown). Some deviations from strict linearity are expected due to assumptions such as ignoring salinity, refreezing, minor interpolation and mapping errors, averaging procedures, and parameterisations that may not scale strictly with $u^\star$. In some models, as described in the methods, $u^\star$ is also inferred from upper-layer velocities rather than directly at the ice-ocean interface, which may further lead to discrepancies in the fits presented here. Further details on model configurations and parameterizations can be found in Table 1.





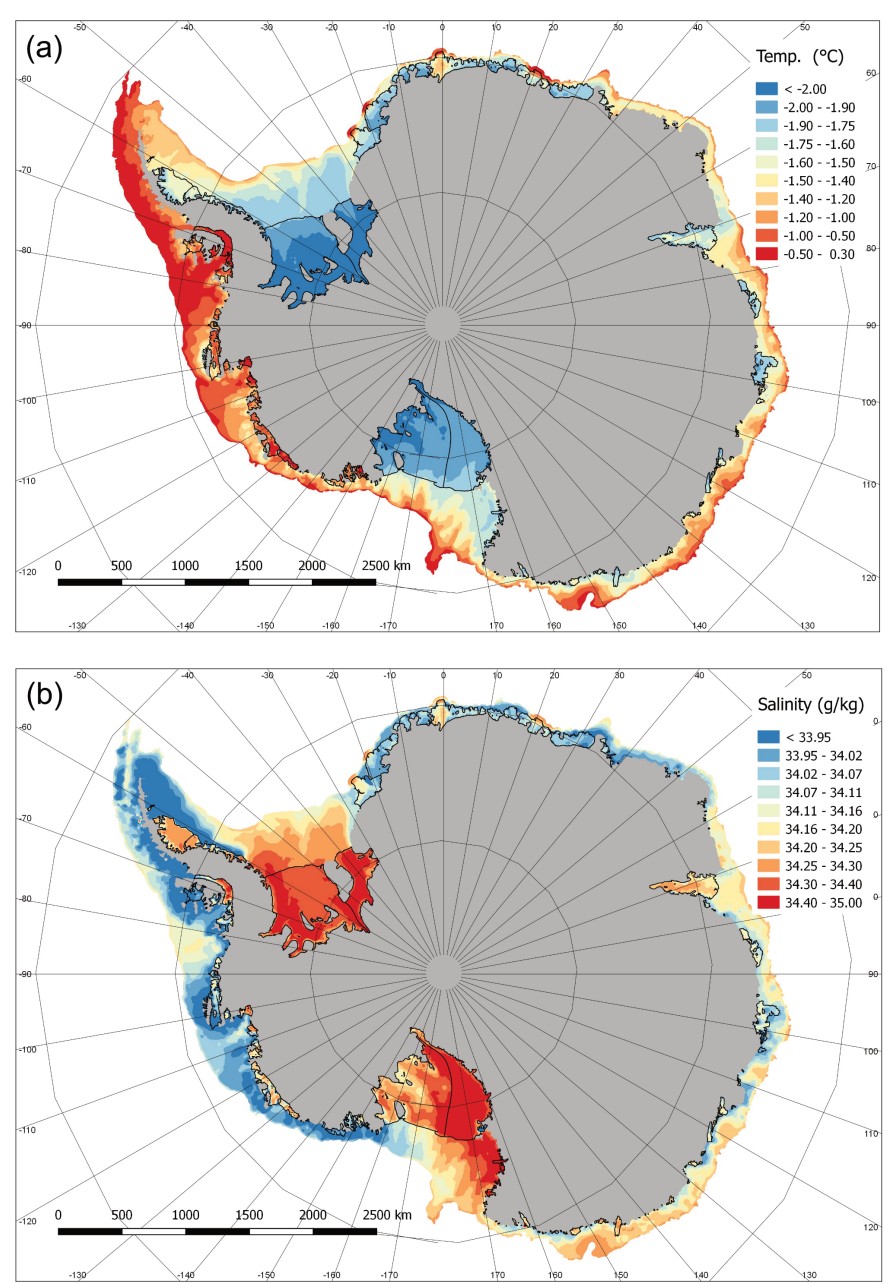

**Figure 4.** Column averaged oceanographic properties for the continental shelf region, including the sub-ice-shelf ocean cavities, for (a) temperature and (b) salinity.





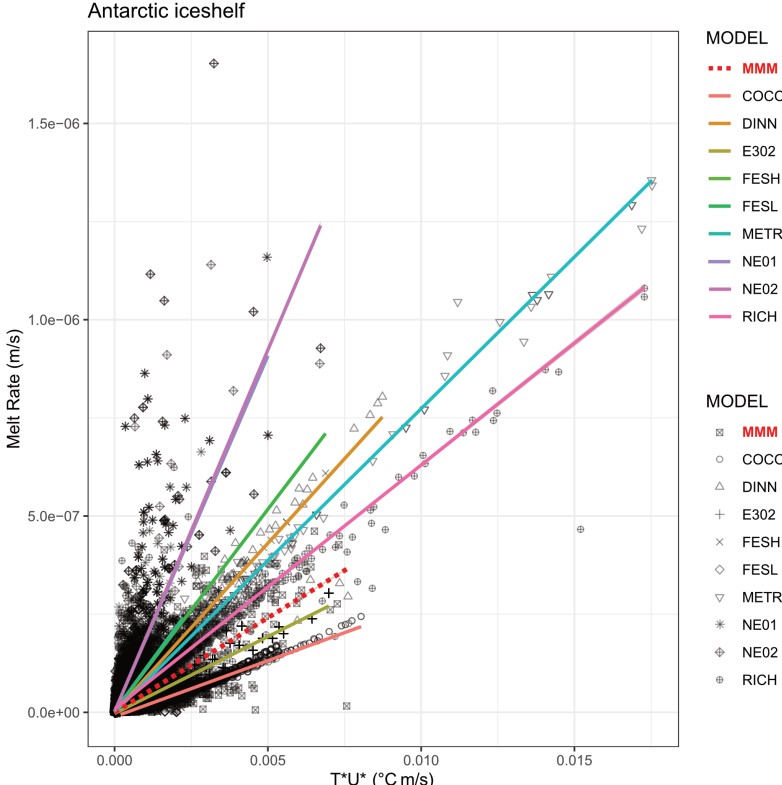

**Figure 5.** The melt sensitivity for each model is shown as the relationship between melting and the product of thermal driving and friction velocity. Solid lines show the linear fit for each model, with the gradient and goodness of fit for each reported in Table 3.

The thermo-kinematic melt sensitivity can be converted to a range of thermal melt sensitivities, $\psi = m/T^\star$ (m year$^{-1}$ °C$^{-1}$), for given values of $u^\star$. We perform this analysis as the thermal melt sensitivity has units that are more intuitively understood as it produces a melt rate per unit of thermal driving, and it is a commonly used approach when parameterising basal melting in ice sheet models. The range of $u^\star$ produced in the MMM has a log-normal distribution ($\mu = -5.9462, \sigma = 0.3283$), which was used to produce estimates of $u^\star$ for one standard deviation spread of the log-normal distribution, giving the lower value of $1.4 \times 10^{-3}$, the mean of $2.4 \times 10^{-3}$, and the upper value of $1.7 \times 10^{-2}$ m s$^{-1}$. From Eqn. 1, these values of $u^\star$ correspond with approximate values of the ocean speed under the ice shelf equal to 2.8, 4.8 and 34 cm s$^{-1}$, respectively, assuming $c_d = 2.5 \times 10^{-3}$ (Table3). For the MMM, the range of melt sensitivities, $\psi$ is 2.13 to 25.86 m °C$^{-1}$ year$^{-1}$, illustrating the important role of the ocean currents and therefore that incorporating the friction velocity is critical as it accounts for turbulent heat transfer to the ice.



**Table 3.** Summary of the melt sensitivity to thermal-driving and kinematic-driving. $\zeta = m(T^\star u^\star)^{-1}$ is the thermo-kinematic melt sensitivity ($°C^{-1}$), taken as the linear slope of the relationship, with corresponding goodness-of-fit - r-squared ($r^2$) - values from Fig.5. Values of the thermal sensitivity $\psi$ (year$^{-1}$°C$^{-1}$) are presented for comparison, using indicative numbers for $u^\star$ for the lower, mean and upper values of the log-normal distribution from the MMM, as described in the text, yielding $1.4 \times 10^{-3}$, $2.4 \times 10^{-3}$, and $1.7 \times 10^{-2}$ m s$^{-1}$, where $\psi = \zeta u^\star \times 31557600$ s year$^{-1}$.

| Identifier | $\zeta$ ($°C^{-1}$) $\times 10^{-5}$ | $\zeta$ ($r^2$) | $\psi$ (m year$^{-1}$°C$^{-1}$)$^{‡}$ | | |
|---|---|---|---|---|---|
| | | | $-1\sigma(u^\star)$ | $\overline{u^\star}$ | $+1\sigma(u^\star)$ |
| MMM | 4.82 | 0.69 | 2.13 | 3.65 | 25.86 |
| COCO | 2.85 | 0.98 | 1.30 | 2.16 | 15.29 |
| DINN | 8.60 | 0.89 | 3.80 | 6.52 | 46.14 |
| E3O2 | 3.92 | 0.96 | 1.73 | 2.97 | 21.03 |
| FESH | 10.3 | 0.80 | 4.55 | 7.80 | 55.26 |
| FESL | 10.7 | 0.72 | 4.73 | 8.10 | 57.40 |
| METR | 7.73 | 0.98 | 3.42 | 5.85 | 41.47 |
| NE01 | 18.1 | 0.68 | 8.00 | 13.71 | 97.10 |
| NE02 | 18.4 | 0.68 | 8.13 | 13.94 | 98.71 |
| RICH | 6.21 | 0.77 | 2.74 | 4.70 | 33.32 |

$^{‡}$ The values of $u^\star$ used are produced by a current speed, $\overline{uv}$, of 2.8, 4.8 and 34 cm s$^{-1}$, respectively, assuming $c_d = 2.5 \times 10^{-3}$ (see Eqn. 1), the approximate average $c_d$ used across all models.

## 4 Discussion

The Multi-Model Mean (MMM) estimate of Antarctic ice shelf basal melting surpasses the performance of any individual model in our ensemble. By integrating multiple state-of-the-art ocean simulations from research teams worldwide, the MMM reduces uncertainties traditionally associated with single-model outputs. This comprehensive approach reveals spatial patterns and regional variations in Antarctic ice shelf melting that are critical for refining projections of future sea-level rise and enhancing our understanding of ice-sheet/ocean interactions. The spatial distributions of melt rates (Figure 1) highlight high melting in the Amundsen and Bellingshausen sectors, consistent with observations of warm oceanic conditions and rapid ice thinning (e.g., Rignot et al., 2013; Shepherd et al., 2018). In contrast, East Antarctica experiences moderate melting in areas such as Cook, Totten, and Fimbul ice shelves. These patterns largely reflect the interplay of warm water pathways, topographic influences, and dynamic ocean currents beneath the ice shelves.

All individual models but one produce lower melt rates than satellite-derived estimates, with the MMM averaging approximately 0.6 m year$^{-1}$ compared to the satellite-based 0.88 m year$^{-1}$. Although E3O2 reached 0.91 m year$^{-1}$, others remain as low as 0.37 m year$^{-1}$, underscoring the substantial spread in simulated melting conditions. Mass loss estimates further highlight these discrepancies: the MMM suggests about 876 Gt year$^{-1}$ from melting and a freeze-to-melt ratio of 3.92%, whereas





satellite products imply 1407 Gt year$^{-1}$ and 18.85%, respectively. These large differences likely stem from variations in spatial
domains used to compute melt rates, the epoch of observational and model data, and potential methodological biases. We also
note the satellite-derived estimates do not include measurements south of 82.4 °S but were extrapolated into these areas where
some of the deepest parts of the Ross and Ronne-Filchner Ice Shelves reside, potentially causing further bias. We suggest the
satellite-derived datasets may overestimate both melting and refreezing, at times yielding physically implausible refreezing
rates that would produce a large value for the freeze-to-melt ratio. However some models (FESOMH and FESOML) due to a
relatively cool ocean, produce about half of the total mass loss (809 and 680 Gt year$^{-1}$), and freeze-to-melt ratio (30.12 %)
that is almost double, as compared with satellite estimates, likely due to a cold-bias in the oceanic conditions (Naughten et al.,
2018).

The models show reasonable consistency in the range of current speeds and temperatures but display considerably larger
variability in salinity distributions. A notable feature is that mean melt rates consistently exceed median values for both the
models and satellite estimates, indicating that a relatively small number of regions with intense melting skew the average
towards higher values. The direct influence of salinity, within the three-equation parameterization, has a weak influence on
basal melting, as compared to thermal and momentum forcings. Although the reasons for the variations in salinity between
models are not clear, differences in open-ocean air and sea–ice fluxes may be a contributing factor. In polar regions, salinity
has a stronger influence on density than temperature, so indirectly, salinity-driven buoyancy circulation may be a leading cause
of the melt variability between models (e.g. Holland et al., 2008a).

The temperature and salinity distributions (Figure 4) around Antarctica's continental shelves highlight the interplay between
off-shelf intrusions of relatively warm, salty Circumpolar Deep Water (CDW) and local Dense Shelf Water (DSW) formation
driven by sea-ice production and air–sea–ice fluxes (e.g., Jacobs et al., 1992; Jenkins and Bombosch, 1991; Rintoul, 2018;
Gwyther et al., 2020a). In West Antarctica, CDW commonly enters the continental shelf via bathymetric troughs, as observed
near Thwaites Glacier, where persistent warm ocean waters lead to high basal melt rates (Shepherd et al., 2018; Rignot et al.,
2019). In contrast, embayments such as Ross and Ronne–Filchner typically remain dominated by colder, saltier DSW, which
can reduce melting by insulating the ice from warmer waters at depth (Hellmer et al., 2012; Timmermann et al., 2012). Al-
though the Amery Ice Shelf is generally regarded as having relatively low melt rates because it is dominated by near-freezing
waters (Galton-Fenzi et al., 2012; Rosevear et al., 2022), our analysis shows most models are producing warmer conditions be-
neath Amery that drive higher melting than expected. These contrasting scenarios underscore the strong influence of localized
polynyas and seasonal processes of brine rejection, meltwater input, and ice formation on water mass modification (Cougnon
et al., 2017; Nakayama et al., 2018).

Freshening signals in areas of intense melting reflect complex feedbacks between melting, circulation, and water mass prop-
erties: glacial meltwater influx lowers salinity and can modify stratification, thereby influencing the spread of warm CDW and
subsequent melting (Jenkins, 2016; Paolo et al., 2018). While CDW intrusions maintain high temperatures and salinities where
they occur, the accompanying freshwater partially offsets these conditions close to the ice shelves and coastlines, emphasizing
that melting is governed by a delicate balance between available heat supply, mixing processes, and salinity-driven density



gradients. These dynamics underscore the strong coupling of physical processes on the continental shelf, and further emphasis how relatively small changes in circulation or sea-ice production can significantly alter local hydrography and basal melt rates.

Our sensitivity analysis underscores the linked relationship between thermal driving and friction velocity in controlling melt rates, that is manifest in the form of the parameterisation. Defining the thermo-kinematic melt sensitivity, $\zeta = m/(T^{\star}u^{\star})$, produces good agreement between the model estimates of melting and the ocean temperature and currents, and the linear fit for each model. The use of $\zeta$ reveals how both heat availability and turbulent mixing govern melting, particularly near grounding zones where $T^{\star}$ and $u^{\star}$ are often roughly equal in their influence on melting. Noting the values we use are those supplied in

the calculation of the melt rate and are close to the ice base and exactly how $T^{\star}$ and $u^{\star}$ are constructed in the sub-ice-shelf cavity from the open ocean remains to be evaluated.

     Our findings confirm that simply considering thermal forcing alone overlooks the significant role of friction velocity in driving basal melt. The range of thermal melt sensitivities derived from this approach illustrates the importance of resolving both temperature and current speed accurately in both ocean models and parameterisations of melting (Burgard et al., 2022; Finu-

cane and Stewart, 2024, e.g.). These insights highlight where improvement in observational data—such as better constraints on cavity circulation and velocity structure—can refine future simulations, thereby enhancing predictions of ice shelf stability and sea-level rise.

## 4.1   Limitations and Assumptions

Despite the advances that may be achieved through multi-model averaging, several limitations and assumptions persist. Models

differ in the areas they use to compute melt rates, especially near grounding zones, and this can bias aggregated results. In our approach, we applied a common spatial framework to all models, which typically underestimates melting compared to calculations over each model's native grid. Consequently, the melt rates reported here are generally lower than what individual model studies produce on their original domains (Table 2).

     The thermo-kinematic sensitivity analysis presented here assumes relative uniformity and may not fully capture phenomena

such as frazil ice formation, complex buoyancy-driven flows, heat flux into the ice shelf, or salinity influences (e.g., Rosevear et al., 2024). We also note that the averaging we use here may also influence the results and ideally estimates of the important parameters would be conducted online, at each model times-step. Unresolved small-scale mixing processes and variability in vertical resolution further complicate melt rate estimates (Gwyther et al., 2020a). Addressing these challenges will require improved domain definitions near grounding zones, refined model resolution and numerics, and parameterization enhancements

(e.g., ocean mixing near ice shelves and frazil ice dynamics).

     Satellite-derived estimates, although indispensable for capturing large-scale patterns of ice-shelf melt, also carry uncertainties. Temporal coverage, resolution constraints, and the challenge of accurately detecting refreezing or thinning beneath thick ice shelves can introduce errors (Paolo et al., 2018; Adusumilli et al., 2020). Field measurements, including deployments of autonomous radar systems (e.g., ApRES) and oceanographic sensors in sub-ice-shelf cavities, are therefore critical for refin-

ing parameterizations and providing ground-truth data (Cook et al., 2022). Closer integration of satellite observations, in situ

 

campaigns, and modeling efforts will help identify and rectify discrepancies such as regions of implausible refreezing rates or underestimates of localized melting.

## 5 Conclusions

This comprehensive multi-model ensemble analysis provides the first Multi-Model Mean (MMM) estimate of Antarctic ice shelf basal melting. As well as providing a useful comparison between different models, this study should prove useful to a range of users, including to help guide the direction of future observations on and beneath ice shelves, thereby integrating ice sheet/ice shelf-ocean observations and modelling.

Our findings highlight the strong dependence of both thermal driving and friction velocity on the melt rate, explicit in the melt parameterisations used in the models, which is especially evident near grounding zones where their combined influence is about equal. The thermo-kinematic sensitivity analysis underscores the sensitivity to incorporating both factors rather than relying solely on thermal forcing, and it highlights where improvements in observational data—such as better constraints on sub-ice-shelf circulation and velocity structure especially adjacent to grounding zones—can refine future simulations.

Although the models collectively under-predict melt rates as compared with satellite-derived estimates, this discrepancy offers insights into sources of potential bias in both remote-sensing products and model parameterisations. Recognizing that a small number of highly active melting regions can skew the mean sheds light on why mean melt rates often exceed median values. In addition, our results underscore the likelihood that some satellite records may overestimate melting and refreezing, making direct comparisons challenging without careful evaluation of spatial domains and data epochs.

Addressing remaining challenges will further reduce uncertainties in ice shelf melt rate projections. These challenges include better capturing refreezing and frazil ice formation, accounting for buoyancy-driven flows, and improving model domains—particularly near grounding zones. Continued integration of satellite-based estimates with in situ observations, such as ApRES and other oceanographic measurements, will help validate sub-ice processes and identify discrepancies in areas with little data.

Ultimately, this multi-model approach advances our ability to project how the Antarctic Ice Sheet will respond to climate change, thereby refining global sea level rise predictions. As numerical models and observational efforts continue to improve, the understanding of the physical processes driving ice shelf melting will become clearer, and projections of Antarctic ice mass loss will gain accuracy and confidence.

*Code and data availability.* Multi-model mean output and derived quantities are available from the Australian Antarctic Division Data Centre: https://data.aad.gov.au/



*Author contributions.* BGF developed and led the project, RPS led the analysis and developed the figures, all authors contributed model
425 results, the analysis and to the writing of the manuscript

*Competing interests.* No competing interests are present

*Acknowledgements.* This research was supported by the Australian Antarctic Program via the Antarctic Gateway Partnership (SR140300001),
the Australian Antarctic Program Partnership (ASCI000002) and the Australian Research Council (ARC) Australian Centre for Excellence
in Antarctic Science (SR200100008). Support was provided by the World Climate Research Programme's Climate & Cryosphere (CliC)
430 core project, the International Union of Geodesy and Geophysics International Association for Cryospheric Sciences (IUGG IACS), and
the Southern Ocean Observing System (SooS). Thanks to Dr Shusheel Adusumilli (University of Oregon, USA) and Prof. Helen Amanda
Fricker (Scripps Institution of Oceanography, USA) for assistance with the satellite data products used here.



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
