# Peer review of "Realistic ice-shelf/ocean state estimates (RISE) of Antarctic basal melting and drivers"

_EGUsphere, 2024_

## Author Comment (AC1)

We thank both reviewers for their comments, which have strengthened our manuscript. Below, we provide a consolidated response to each reviewer's feedback, with our replies in blue following each of their comments.

**RC1: 'Comment on egusphere-2024-4047', Anonymous Referee #1, 20 Mar 2025**

The authors present the multi-model mean (MMM) of ice shelf melt rates and other parameters that determine ice shelf melt rates. They claim that the MMM provides a useful comparison between different models and serves as a guideline for observations and modeling. However, I believe that the simulated ice shelf melt rate is a parameter that can be easily tuned by selecting appropriate coefficients, and the multi-model comparisons presented here are somewhat overstating and misleading. I believe more analyses can make this manuscript much more useful for the community. I suggest a major revision.

We note that each of the points raised in the reviewer's preamble are also again repeated below, where we provide our response. However, we use this opportunity to clarify the rationale behind our use of the multi-model mean (MMM) for ice shelf melt rates and have included this additional text in the supplemental information that we refer to in the main document.

"Each numerical model has a distribution that represents its prediction of basal ice melt. Due to the complexity of the underlying physical processes, individual models inherently possess distinct biases and uncertainties reflected in the spread and shape of their respective distributions. Consequently, some models may overestimate or underestimate the melt rates or their spatial variability. However, when these models are averaged, the central tendency of the ensemble tends to converge toward a more robust estimate of the true state. This behaviour is consistent with statistical theory, particularly the 'central limit theorem', where the average of multiple independent estimators tends toward a Gaussian distribution, even if individual models are not normally distributed. The ensemble mean thus serves to reduce individual biases and smooth out uncertainties, resulting in a more reliable representation of basal melt. This "Gaussian phenomenon" of multi-model averaging is well documented in the climate modelling literature. For example: Tebaldi & Knutti (2007) highlight the statistical benefits of ensemble averaging for climate projections, particularly in reducing bias and quantifying uncertainty;  Gleckler et al. (2008) show that multi-model means often outperform individual models based on  performance metrics;  Smith et al. (2009) provide a Bayesian framework that supports the Gaussian nature of multi-model means and their improved predictive accuracy; Knutti & Sedláček (2013) emphasise how multi-model means enhance robustness and reliability in CMIP5 projections. The MMM represents a useful benchmark for assessing model diversity and guiding observational design and model development."

References

Gleckler, P. J., Taylor, K. E., and Doutriaux, C.: Performance metrics for climate models, Journal of Geophysical Research: Atmospheres, 113, 2008.

Knutti, R. and Sedláček, J.: Robustness and uncertainties in the new CMIP5 climate model projections, Nature climate change, 3, 369-373, 2013.

Smith, R. L., Tebaldi, C., Nychka, D., and Mearns, L. O.: Bayesian modeling of uncertainty in ensembles of climate models, Journal of the American Statistical Association, 104, 97-116, 2009.

Tebaldi, C. and Knutti, R.: The use of the multi-model ensemble in probabilistic climate projections, Philosophical Transactions of the Royal Society A: Mathematical, Physical and Engineering Sciences, 365, 2053-2075, 2007.

**Major Comments**

1. The authors discuss, for example, the comparison of simulated total melt with satellite-based estimates (e.g., between lines 325 and 337 and in Table 2). However, in global simulations like this, integrated melt rates are easily tunable parameters. I question the significance of this study's comparison of model outputs that are easily adjustable, rather than focusing on parameters that are more challenging to tune and simulate. I suggest that the authors add a paragraph explaining how the drag coefficients and heat and salt transfer coefficients are determined in all simulations.

The reviewer's comments suggest the melt rates are easily tunable, which they are not. If the models were so easy to tune then many of the models used here would show a mean melt rate that was comparable with the observations, whereas in fact they do not, and exhibit a large spread from each other and the observations in both magnitude and spatial patterns.

Furthemore, tuning of melt rates is particularly problematic if the coefficients are global or circum-Antarctic, as in all the simulations used here. Several published examples do show for individual ice shelves that some adjustment of different parameters, such as the friction coefficient, the exchange coefficients, or vertical mixing coefficients (depending on how the melt rates are parameterised) can produce a more reasonable integrated melt rate, but often the spatial pattern is inconsistent as compared with observations. This practice, however, is oftentimes not robust and the interdependence of each of the parameters means establishing the actual controlling mechanisms on melt rates is difficult. We have added a sentence to the manuscript in the methods section to highlight this fact, as follows:

"Note that each model employs globally constant parameters and equations for all ice shelves across its domain."

The ice shelf melt rate is parameterised in the same way for each model, except COCO, as we describe, and is dependent on both the speed and temperature of the ocean current underneath the ice shelf. The melt rate is therefore the end result of many complex oceanic processes, and relationships and feedbacks with those and the shape of the geometry and forcing, that provide for the resulting ocean heat supply underneath the ice shelf. The integrated effect that the melt rates therefore show is a key reason as to why we decided to focus the analysis on the melt rates.

The determination of the heat and salt transfer coefficients are already explained in a section explicitly devoted to this (see Section 2.1), including information required to support the differences between models in Table 1.

2. To enhance the usefulness of the comparisons, I suggest addressing the following aspects (a-d). In the current version of the manuscript, the authors discuss bottom or vertically integrated ocean hydrography, which is not the best metric for determining ice shelf melt rates. I recommend presenting thermocline depths and vertical sections under major ice shelf cavities to illustrate how variations in hydrography lead to differences in simulated melt rates. Additionally, it is important to examine interannual and seasonal variability, discussing the extent of differences in these aspects.

(a) Integrated ice shelf melt rates for each individual ice shelf.

The integrated rates for each of the seven ice shelves we selected for the study and for each model, including the Multi-Model Mean and observations, are shown in the supplemental information, and a subset been moved to a table in the manuscript that includes columns of the average temperature and salinity on the continental shelf in the front of each of the seven ice shelves selected for the study, and all of Antarctica.

(b) Hydrographic conditions, particularly vertical sections within the ice shelf cavity or at the ice shelf front.

(c) Temporal variability of thermocline depth and its relationship to ice shelf melt rates.

(d) The processes governing ice shelf melt rates, such as thermocline depth variations, temperature changes, and other factors influencing ocean velocity.

These are not possible to calculate across all ice shelves, as not all fields were provided from each modelling group. We have focused this paper on understanding the differences in simulated melt rates. We think that even if all fields (including the full suite of hydrographic conditions) are available, which they are not, their inclusion would also change the focus of the paper from what we have intended. We agree future studies should indeed focus on these aspects, potentially finding our study useful to motivate those efforts.

**Minor Comments**

1. I find that Antarctica's wide mean melt rate is not very intuitive. I suggest adding an integrated melt rate in Gt/yr.

These values are reported in many locations in the paper, including the abstract, Table 2 and throughout the text and, as described above, these are also available in both the supplemental information for each of the seven key ice shelves we used in the study.

2. Lines 79-82: Considering that the Antarctic ice shelf melt rate is an easily tuned parameter, I feel this is somewhat overstating the presented findings here "allows us to constrain both present and future ocean-driven impacts on Antarctica, and provide much needed evaluation with both satellite-derived and in situ estimates of ice shelf basal melt rates".

This comment has been addressed under Major Comment 1. In regard to this specific sentence we have amended it to now say "Comparing these results allows us to **better** constrain both present and future ocean-driven impacts on Antarctica, and provide much needed evaluation with both satellite-derived and in situ estimates of ice shelf basal melt rates."

3. Lines 403–412: One of the main conclusions of this multi-model comparison is the strong dependence of both thermal driving and friction velocity on the melt rate—concepts that have already been well established in numerous previous studies. While this study represents a first attempt at presenting a multi-model mean, I do not believe that there are more useful comparisons metrics that are not presented in the current version of the manuscript.

We were not suggesting that the formulations were not already well established. In fact, as they are well established and form the basis of the models, the approach represents the appropriate method of comparing the models. To our knowledge, no study has provided continental scale comparisons of these parameters and across models, and with observations, as we have done. Therefore, this manuscript presents a useful and novel study that should be of broad interest to a range of research areas.

**RC2: 'Comment on egusphere-2024-4047', Anonymous Referee #2, 04 Apr 2025**

**General Comments**

This study compares several circum-Antarctic ocean simulations that include sub-ice-shelf cavities to highlight inter-model differences and evaluate results against satellite-derived melt rates. The approach is to take the ocean simulations "as is," without following a particular protocol. Both model outputs and satellite-derived melt rates are remapped to a common grid to facilitate comparison. This represents the first attempt to provide a Multi-Model Mean estimate of Antarctic ice shelf basal melting, while also

highlighting areas of agreement and divergence among models. The work has the potential to serve as a valuable reference for future modeling efforts and to help guide targeted observational strategies.

However, I have a major concern that the comparison is overly focused on melt rate and its primary drivers (friction velocity and thermal driving) while giving limited or no attention to the underlying ocean and sea ice state. These parameters are often highly tunable, which reduces the robustness and generalizability of the findings. As a result, the comparison has limited utility for informing future modeling or observation-based studies. A broader focus on key oceanographic and sea ice metrics would strengthen the study's relevance and impact.

We have addressed a similar point in response to Reviewer 1. The reason the melt rate was chosen as the primary metric for evaluation is that it was readily available across all models, and represented the integrated influence of all other oceanic and forcing processes that conflate to produce the melt rate. We feel the additional analysis, given the length of our current manuscript, its focus on the melt rates, and unavailability of some of the fields needed to perform the analysis, make the work proposed by Reviewer #2 outside of the scope of the present paper.

Here are several suggested comparisons that I believe would significantly strengthen the manuscript. While I do not expect the authors to include all of these, incorporating at least some would enhance the study's depth and utility:

1. Cross-sections (y–z) of mean temperature and salinity in key regions, particularly where Circumpolar Deep Water intrusions occur;
2. T/S diagrams over individual ice shelves, including adjacent continental shelves;
3. Heat transport across individual ice shelf fronts;
4. Mean mixed layer depth (e.g., July to September), based on a clearly defined criterion;
5. Annual cycle of sea ice area and volume;
6. Maps of sea ice concentration and sea ice thickness.

For metrics where observation-based estimates are available (e.g., sea ice properties), I encourage the authors to include these in the comparisons as well.

These are valuable ideas for analysis but the data management makes it unworkable for the current paper. Importantly, not all variables for this generation of models are available and so comparison of these additional variables is not possible. For example, not all outputs are available and not all of these models include a sea ice sub-model, limiting us from making this comparison. As such this additional work is not possible for our study. We agree that a more in-depth study would be useful and our hope is our paper would provide motivation for further and subsequent model intercomparisons that are perhaps more tightly controlled. However, this study is focussed on using basal melting as its primary focus.

The manuscript is overall well-written and clear. However, in addition to my main concern noted above, there are also significant issues with several figures, as outlined below. For these reasons, my recommendation is major revision.

**Specific Comments**

Table 1: I suggest including (1) a column indicating whether the model is "regional" or "global"; (2) a column specifying the atmospheric forcing used to drive the model (e.g., JRA-55, CORE); and (3) for regional models, a column identifying the product used to drive open boundary conditions.

This information is available in the supplementary material . The products used for the lateral ocean boundary conditions of the regional models have been added.

Lines 142–143: "The thermal driving…" – Please modify this sentence to reflect that thermal driving can be negative (i.e., T* < 0), in which case the water is colder than the freezing point.

Modified to say both melting and freezing.

Section 2.2 "Remapping approach to a common grid":

I was not familiar with the Voronoi tessellation approach, which I believe is not commonly used in the geophysical sciences. What are the advantages of this method compared to more widely used techniques such as bilinear or nearest-neighbor interpolation? It is mentioned that this approach results in "lower total melt estimates than previous studies" (lines 199–200), and later that "discrepancies likely arise from differences in the areas used to compute melt rates" (lines 252–253). Is this because the Voronoi tessellation is non-conservative? Would it make more sense to convert melt rates to a mass flux, apply a conservative remapping to the common grid, and then convert the flux back to a melt rate? This might avoid the bias toward lower melt estimates. Lastly, please include other specific reasons for selecting this approach over more conventional alternatives.

We have added additional information to the supplemental information, as follows: "

Whilst the Voronoi tessellation approach is less commonly applied in geophysical sciences compared to interpolation methods like bilinear or nearest neighbour, it offers several distinct advantages that motivated our selection. Firstly, Voronoi tessellation avoids artificial smoothing by assigning each output cell the value of the nearest input point. This contrasts with bilinear interpolation, which blends surrounding values and may introduce artifacts—especially near boundaries or steep spatial gradients. The method was adaptive to the use of several model outputs which all have different, and irregular meshes and avoids the extrapolation bias inherent in nearest-neighbour and bilinear methods, which can produce misleading values in sparsely sampled or irregularly distributed datasets (See Figure S1). The Voronoi approach inherently adapts to the spatial configuration of the input data, providing a natural partitioning of space that reflects data density. Lastly, it is reproducible and parameter-free. With no tunable parameters this method is deterministic and does not rely on user-defined parameters, ensuring consistency and reduced subjectivity in the data processing. These characteristics made Voronoi tessellation particularly suitable for our use in this case, where spatial heterogeneity and edge preservation were critical in comparing several model outputs. See \citet{Okabe_2000} and \cite{Shepard_1968} for more information."

The observed differences in total melt estimates are not due to the non-conservative nature of the Voronoi tessellation, but rather due to differences in the spatial coverage (i.e., masks) of melt rates provided by each model. The lower melt rates than previous studies are due to remapping these estimates onto a common spatial mapping. Each contributed model defines its own ice shelf mask, which determines the spatial extent over which melt rates are computed. When compiling and comparing model outputs, these slight variations in mask boundaries lead to differences in the integrated multi-model mean - this resulted in slightly lower total melt value, as some models reported. This masking issue is reflected in the literature e.g., (Jourdain et. al., 2020, Nowicki et al., 2016; Rignot, et al., 2013).

References

Jourdain, N. C., et al. (2020). A protocol for calculating basal melt rates in the ISMIP6 Antarctica projections. The Cryosphere, 14, 3111–3134. https://doi.org/10.5194/tc-14-3111-2020

Nowicki, S. M. J., et al. (2016). Ice Sheet Model Intercomparison Project (ISMIP6) contribution to CMIP6. Geoscientific Model Development, 9, 4521–4545. https://doi.org/10.5194/gmd-9-4521-2016

Rignot, E., et al. (2013). Ice-shelf melting around Antarctica. Science, 341(6143), 266–270. https://doi.org/10.1126/science.1235798

Okabe, A., Boots, B., Sugihara, K., & Chiu, S. N. (2000). Spatial Tessellations: Concepts and Applications of Voronoi Diagrams (2nd ed.). Wiley.

Shepard, D. (1968). A two-dimensional interpolation function for irregularly-spaced data. Proceedings of the 1968 23rd ACM National Conference, 517–524. https://doi.org/10.1145/800186.810616

Table 2: Are the mean melt rates area-weighted? If so, please clarify this in both the table caption and the main text.

The mean melt rates are the spatial averages. Clarification has been added to the text.

Figure 1: I suggest that the comparison with satellite observations be moved from the supplementary material to the main text.

In the process of writing the document we did discuss this idea but wanted to present the results of the multi-model mean as a priority figure. The evaluation with the satellite data represents an evaluation step, and we remain committed to the presentation of those results in the supplementary material.

Lines 316–317: "The Multi-Model Mean (MMM) estimate of Antarctic ice shelf basal melting surpasses the performance of any individual model in our ensemble." The rationale for this statement is unclear. No analysis is provided to support it. Please clarify the basis for this claim or include supporting evidence.

This paragraph containing this sentence has been amended to be more explicit with our meaning of what we think the MMM surpasses the performance of any individual model in our ensemble, with the following text included:

"By integrating multiple state-of-the-art ocean simulations from research teams worldwide, the MMM reduces uncertainties traditionally associated with single-model outputs. This approach is consistent with the 'central limit theorem', where the ensemble average of the models tends to a Gaussian distribution, even if individual models are not normally distributed. The MMM thus serves to reduce individual biases and smooth out uncertainties, likely resulting in a more reliable representation of the State Estimate of basal melting and ocean-drivers. The Multi-Model Mean (MMM) estimate of Antarctic ice shelf basal melting therefore likely surpasses the performance of any individual model in our ensemble. "

Line 399: "Comprehensive" may be too strong in this context. I suggest using a softer term such as "extensive."

Agreed and "comprehensive" has been deleted without replacement.

Code and Data Availability: Please make the code used in this study publicly available to enhance transparency and reproducibility.

Yes - that is certainly our intention. Code and data will be made available through the Australian Antarctic Division Data Centre

**Editorial / Typographical Comments**

Line 55: "IUGG, WCRP, SCAR" – Please define these acronyms in full for readers who may not be familiar with them.

Changed.

Line 60: Please define MISOMIP and MISOMIP2.

Changed.

Line 83: "...the project" – Since the RISE Project is introduced in the next paragraph, I suggest changing "the project" to "this work" or a more neutral phrasing.

Changed.

Lines 90–91: Please define ICESat, ApRES, NECKLACE, and SOOS.

Changed, except for defining NECKLACE which is defined in the context of the sentence.

Table 1: The phrase "...reference for each model" is redundant. I suggest deleting "for each model." Also, please define u* in the caption

Deleted and u* is now defined in the caption.

Lines 149–150: "Thermal driving …" – This is unnecessarily repetitive, as thermal driving was already defined in line 142. I recommend combining the two definitions to avoid duplication.

Fixed. Thanks for picking up that repetition.

Line 228: Correct "at teh" → "at the."

Fixed.

Figure 1: Please define "ISMR" and "SD." The coordinates are difficult to read; consider increasing the font size of the x and y axes.

Defined and enlarged.

Figures 2 and 3: Same comments as for Figure 1.

Fixed.

Line 261: Correct "is suggests" → "suggests."

Fixed.

Line 375: Fix formatting error in "e.g.,[" – likely a LaTeX code issue.

Fixed.

Supplementary Material: In some figures, the legend text is difficult to read. Please ensure all text is legible at standard PDF viewing size.

Enlarged.

---

## Author Response (AR1)

Dear Nico,

Thanks for your comments and response. Our responses to your comments are in blue following each of yours.

Dear Ben Galton-Fenzi and co-authors,

First of all, I thank the two experts for their careful review of the initial manuscript. I also thank you for your point-by-point responses.

Both reviewers recommended to reconsider the manuscript after major revisions. Their major concerns are related to (i) the absence of analyses of temperature and salinity in front of the ice shelf cavities, and (ii) the lack of evidence for the claim that such a multi-model mean is more interesting than individual models.

On the first point, more analyses and disussions about the T,S properties in front of ice shelves is needed. Melt rates in regions with warm cavities can be adjusted by tuning the heat exchange coefficient (Jourdain et al., 2017), even though such adjustment is indeed more difficult for circum-Antarctic simulations. Regardless of the tuning, both reviewers find that the study would be much more useful if it included more assessment of temperature and salinity. Looking at the multi-model mean temperature and salinity at a given depth (or profiles averaged over a given region) would indicate whether biases are actually weak in the multi-model mean, and whether realistic melt rates are obtained for realistic conditions near the ice shelf fronts. The new table in the manuscript that will include "the average temperature and salinity on the continental shelf in the front of each of the seven ice shelves" may partly address this concern if it is enough described and discussed.

We have included averaging and the standard deviation for temperature, salinity for the cavities of the seven ice shelves and the open ocean on the continental shelf in their vicinity, included in Table 3 and linked to the discussion.

Regarding the second point, Reviewer #1 was not convinced by the interest of the multi-model mean. I consider that the use of the multi-model mean is well justified and is an interesting aspect of this manuscript. However, the provided references are related to climate models under a common forcing framework (CMIP), while the models partaking to RISE were run under very different forcings. In particular, I am not sure that averaging ocean models forced by reanalyses over the recent period together with a coupled ocean–atmosphere model in pre-industrial conditions can be expected to reduce the overall bias. I therefore expect a slightly more nuanced presentation of the added value of a multi-model mean in the revised manuscript.

We appreciate the editor's and reviewer's thoughtful comments on the use of the multi-model mean (MMM). We agree that unlike in coordinated modelling frameworks such as CMIP, the models participating in RISE were not all forced under identical or standardised conditions. As such, the ensemble models reflect not only structural differences in model physics and numerics, but also variations in boundary forcing and resolution. We acknowledge that this heterogeneity limits the strict applicability of ensemble theory in its classical statistical sense - particularly the assumption that model errors are independent and identically distributed. Therefore, while the 'central limit theorem' provides a conceptual basis for understanding ensemble averaging, its assumptions are only partially met in the RISE ensemble. Nonetheless, we argue that the MMM retains utility as a descriptive metric that synthesises diverse model behaviour, especially when interpreted with appropriate caution. In this context, the MMM does not necessarily represent an optimal or unbiased estimate of the true melt rate but provides a 'first-order synthesis' of model diversity. It can help identify robust spatial patterns and highlight areas of persistent inter-model disagreement. As advised, we have updated the manuscript under the Assumptions and Limitations section in the discussion,

to reflect this more nuanced interpretation and explicitly discuss the caveats associated with averaging models forced under different conditions.

In the revised version, please also provide an explicit link to the data and code shared through the AAD data center.

This is not available to us at this time as the data are embargoed until publication proceeds but we can provide the explicit link once it has been provided to us.

Regards, Ben Galton-Fenzi on behalf of co-authors.

---

## Author Response (AR2)

Dear Nico,

Thanks again for your comments and your balanced approach to handling our manuscript. We also thank the reviewers for their thoughtful and engaged review of our manuscript. Our responses to your comments are in blue following each of yours.

Dear Ben Galton-Fenzi and co-authors,

First of all, I thank the two reviewers for their second evaluation of your manuscript. Reviewer #1 recommends accepting your manuscript as it is. In contrast, Reviewer #2 still thinks that the manuscript is problematic and that there were more interesting things to analyse in this model ensemble than the multi-model mean melt rate. I doubt that you will be willing to follow this direction and entirely change the content of your manuscript. Based on my own review and Reviewer #1's evaluation, I do think that there is value in presenting the RISE results through the multi-model mean melt rate. Several points raised by Reviewer #2 are nonetheless very relevant, and I am asking you to revise your manuscript as detailed below.

1- "realistic ice-shelf/ocean state estimate [...]" may not be the most appropriate title, as (i) "realistic state estimate" is usually dedicated to ocean simulations with some kind of data assimilation, which is quite far from this ensemble that even includes a pre-industrial simulation, and (ii) "Realistic" is a claim that is not supported by a thorough evaluation of the model outputs in comparison to observations (e.g., 3D temperature and salinity). As such, the title of this manuscript and the name of the RISE project may sound somewhat problematic. Please consider changing the title to something like "Ice-shelf/ocean multi-model state estimates of Antarctic basal melting and drivers"? Then you can still mention the RISE project in the abstract and/or introduction because this name has been widely advertised in the community.

We have considered your request and agree the title was inappropriate. The new title of the manuscript is

"Multi-model Estimate of Antarctic Ice-Shelf Basal Mass Budget and Ocean Drivers"

2- These sentences in section 4 are still overselling the method (see Reviewer #2's comments and my previous review): "The MMM thus serves to reduce individual biases and smooth out uncertainties, likely resulting in a more reliable representation of the State Estimate of basal melting and ocean-drivers. The Multi-Model Mean (MMM) estimate of Antarctic ice shelf basal melting therefore likely surpasses the performance of any individual model in our ensemble". Please specify that only a part of the uncertainty can be expected to be removed here. The structural uncertainty shared by models, like an inappropriate physical parameterisation of ice-shelf basal melting or common biases in the bathymetry or ice topography, will of course remain problematic in the MMM. Furthermore, the last sentence of the quote should be either removed or supported by analyses as there could still be a few models that are better tuned than the others and that therefore surpass the MMM.

The last sentence has been removed, and additional information has been added to clarify the uncertainty inherent in the MMM, as follows.

The text has been changed from:

"The MMM thus serves to reduce individual biases and smooth out uncertainties, likely resulting in a more reliable representation of the State Estimate of basal melting and ocean-drivers. The Multi-Model Mean

(MMM) estimate of Antarctic ice shelf basal melting therefore likely surpasses the performance of any individual model in our ensemble."

 to:

"The MMM thus serves to reduce biases and smooth out uncertainties, likely resulting in a more reliable representation of basal melting and its ocean-drivers. We note, however, that the uncertainties in the boundary conditions or in the numerical methods and parameterisations shared by models, such as the use of common parameterisations of ice-shelf basal melting or common biases in the geometry, will still contribute to the uncertainty in the MMM."

We have also removed the following sentence from the manuscript: "The Multi-Model Mean (MMM) estimate of Antarctic ice shelf basal melting therefore likely surpasses the performance of any individual model in our ensemble."

3- In Tab.1, add a footnote near the "150 years" of E302 to indicate "pre-industrial conditions" (currently, it is hidden in the supplementary material, and this appears to me as an important piece of information).

Added.

Regards, Ben Galton-Fenzi on behalf of co-authors.